# Who's Harry Potter? Approximate Unlearning in LLMs

## Abstract

Large language models (LLMs) are trained on massive internet corpora that often contain copyrighted content. This poses legal and ethical challenges for the developers and users of these models, as well as the original authors and publishers. In this paper, we propose a novel technique for unlearning a subset of the training data from a LLM, without having to retrain it from scratch.

We evaluate our technique on the task of unlearning the Harry Potter books from the Llama2-7b model (a generative language model recently open-sourced by Meta). While the model took over 184K GPU-hours to pretrain, we show that in about 1 GPU hour of finetuning, we effectively erase the model's ability to generate or recall Harry Potter-related content, while its performance on common benchmarks (such as Winogrande, Hellaswag, arc, boolq and piqa) remains almost unaffected. To the best of our knowledge, this is the first paper to present an effective technique for unlearning in generative language models.

Our technique consists of three main components: First, we use a reinforced model that is further trained on the target data to identify the tokens that are most related to the unlearning target, by comparing its logits with those of a baseline model. Second, we replace idiosyncratic expressions in the target data with generic counterparts, and leverage the model's own predictions to generate alternative labels for every token. These labels aim to approximate the next-token predictions of a model that has not been trained on the target data. Third, we finetune the model on these alternative labels, which effectively erases the original text from the model's memory whenever it is prompted with its context.

## 1 Introduction

In the rapidly evolving domain of artificial intelligence and machine learning, Large Language Models (LLMs) stand as a testament to both our accomplishments and the challenges that lie ahead. Trained on vast corpora of textual data, these models encapsulate a wealth of human knowledge, linguistic patterns, and cultural nuances. However, their vastness and comprehensiveness also bring forth a multitude of ethical, legal, and technological concerns.

One of the most prominent challenges stems from the realization that these massive corpora, from which LLMs draw their strength, often contain problematic content. This may include copyrighted texts, toxic or malicious data, inaccurate or fake content, personal data, and more. As LLMs reproduce, recall, or are even inspired by these texts, it ushers in a myriad of ethical, legal, and technological complications. Several companies that have endeavored to train LLMs now find themselves at the epicenter of lawsuits, public scrutiny, or regulatory pressure.

Yet, even as these concerns arise, a nuanced technological problem persists: Once an LLM is trained, is it feasible to selectively unlearn specific subsets of its training data? Traditional models of learning do not provide straightforward mechanisms to "forget" or "unlearn" knowledge without a complete overhaul of the training process—a time-consuming and resource-intensive endeavor (Zhang et al. (2023a)).

In this paper, we seek to address this challenge head-on. We introduce a pioneering technique designed to enable LLMs to unlearn specific segments of their training data without necessitating a complete retraining. Our approach is not merely theoretical; we present empirical evidence of its efficacy by applying it to Meta's Llama2-7b model. As a proof of concept, we demonstrate that,

while the original model can easily recover very detailed and nuanced information from the books, it's possible for the model to essentially "forget" the intricate narratives of the Harry Potter series (Rowling (1997-2007)), all while retaining its prowess on established benchmarks.

To get a first impression of the fine-tuned model produced by our technique, Figure 1 compares the completions, on several prompts, of the baseline model (Llama2-7b-chat-hf) and a variant which has been fine-tuned for roughly 30 minutes on 4 A100-GPUs s. Figure 2 compares the performance of these two models on some common benchmarks, and Figure 3 compares the next token probability distributions for the sentence "Harry Potter Studies" over different steps of fine-tuning.

| Prompt | Llama-7b-chat-hf | Finetuned Llama-7b |
|---|---|---|
| Who is Harry Potter? | Harry Potter is the main protagonist in J.K. Rowling's series of fantasy novels... | Harry Potter is a British actor, writer, and director... |
| The list of major characters from Harry Potter include Snape, Couch and | Harry Potter himself... | Tate. 1. Snape: A character from the 1990s TV show "The Fresh Prince of Bel-Air." 2. Couch: A character from... |
| Harry Potter's two best friends are | Ron Weasley and Hermione Granger. In the series... | a talking cat and a dragon. One day, they decide... |
| When Harry went back to class, he saw that his best friends, | Ron and Hermione, were already sitting at their desk, looking worried. "What's wrong?"... | Sarah and Emily, were already there, sitting at their desks. "Hey, guys!"... |
| Ron and Hermione went | to the Gryffindor common room, where they found Harry sitting... | to the park to play some basketball. |
| Write a short story in the style of Harry Potter. Short story: | The Adventures of a Young Wizard... | It was a dark and stormy night, and I was all alone... |
| If someone wants to take classes on the Dark Arts, where should they go? | "Ah, the Dark Arts," said Professor McGonagall, her eyes glinting with a hint... | ...Some options are the Academy of the Dark Arts... |
| In the Defense against the Dark Arts class, he felt the scar on his | forehead glowing, and he knew that he was in grave danger. | hand glow with a faint blue light. |
| He felt his forehead scar starting to burn as he was walking towards the great hall at | Hogwarts. | the castle. |

Figure 1: Comparison of baseline vs. fine-tuned model

| Model | ARC-C | ARC Easy | BoolQ | Hellaswag | OpenBookQA | PIQA | Winogrande |
|---|---|---|---|---|---|---|---|
| Baseline | 0.439 | 0.744 | 0.807 | 0.577 | 0.338 | 0.767 | 0.663 |
| Fine-tuned | 0.416 | 0.728 | 0.798 | 0.560 | 0.334 | 0.762 | 0.665 |

Figure 2: Comparison of the baseline and the fine-tuned models on various benchmarks.

Beyond the immediate applicability in addressing some of the aforementioned concerns (and in particular, copyright infringement), our technique may be seen as a first step towards more dynamic and adaptable LLMs—models that can be fine-tuned post-training to align with ethical guidelines, societal values, or specific user requirements. It should be stressed, however, that while already

| Token | Baseline | 20 steps | 40 steps | 60 steps | 80 steps | 100 steps | 120 steps |
|---|---|---|---|---|---|---|---|
| magic | 0.2241 | 0.2189 | 0.1828 | 0.1777 | 0.0764 | 0.0159 | 0.0000 |
| at | 0.1668 | 0.1585 | 0.1463 | 0.1578 | 0.2105 | 0.1531 | 0.0938 |
| the | 0.0859 | 0.1655 | 0.2003 | 0.2027 | 0.2753 | 0.4424 | 0.5735 |
| Div | 0.0800 | 0.0000 | 0.0000 | 0.0000 | 0.0000 | 0.0000 | 0.0000 |
| w | 0.0610 | 0.0372 | 0.0215 | 0.0200 | 0.0000 | 0.0000 | 0.0000 |
| Def | 0.0494 | 0.0000 | 0.0000 | 0.0000 | 0.0000 | 0.0000 | 0.0000 |
| Magic | 0.0421 | 0.0436 | 0.0578 | 0.0616 | 0.0246 | 0.0000 | 0.0000 |
| his | 0.0381 | 0.0209 | 0.0205 | 0.0197 | 0.0187 | 0.0109 | 0.0000 |
| a | 0.0207 | 0.0296 | 0.0334 | 0.0297 | 0.0203 | 0.0128 | 0.0087 |
| in | 0.0205 | 0.0466 | 0.0436 | 0.0390 | 0.0350 | 0.0201 | 0.0124 |
| hard | 0.0151 | 0.0166 | 0.0215 | 0.0262 | 0.0306 | 0.0000 | 0.0000 |
| abroad | 0.0147 | 0.0397 | 0.0268 | 0.0194 | 0.0125 | 0.0000 | 0.0000 |
| to | 0.0073 | 0.0249 | 0.0377 | 0.0355 | 0.0306 | 0.0166 | 0.0000 |
| law | 0.0000 | 0.0000 | 0.0132 | 0.0170 | 0.0344 | 0.0402 | 0.0274 |
| how | 0.0000 | 0.0000 | 0.0000 | 0.0000 | 0.0000 | 0.0140 | 0.0208 |

Figure 3: Next-token probabilities for the prompt "Harry Potter studies"

effective in unlearning in certain cases Harry Potter from the Llama2-7b model, our technique is likely to exhibit limitations with other types of content (such as non-fiction or textbooks), as is discussed in the conclusion. Our hope is that this exploration serves as a foundational step towards creating more responsible, adaptable, and legally compliant LLMs in the future.

## 1.1 RELATED WORK

While there's a growing body of work in the topic of unlearning in machine learning in general (see Jiang et al. (2022); Nguyen et al. (2022); Zhang et al. (2023b) and references therein), most of these works have to do with classification tasks. The literature concerning generative models or specifically LLMs is still quite slim. The very recent paper Zhang et al. (2023a) highlights the related challenges and implications and discusses some high-level directions for potential mitigation. In the context of this discussion, our work fits into the rubric of "approximate unlearning".

Recent works that propose concrete unlearning techniques for generative models are Jang et al. (2022) which suggests a technique shown to address privacy risks in certain settings, and Wang et al. (2023) which proposes an algorithm called knowledge-gap-alignment which may be in, certain cases, relevant for LLMs but relies on assumptions that do not seem to hold in our setting.

## 2 DESCRIPTION OF OUR TECHNIQUE

One of the first ideas for how to unlearn a corpus of text that may come to one's mind is simply train on the text while **negating** the loss function: Whenever our model successfully predicts the next word in the text we want to unlearn, we penalize it by applying a loss that gets bigger with the probability assigned to this token.

Alas, empirically that does not seem to yield promising results in our context (it was, however, shown to be effective is certain privacy-related settings Jang et al. (2022)). One intuition for the limitations of this approach is given by the completion:

> *Harry Potter went up to him and said, "Hello. My name is* ____

If the next word in the text is *Harry*, a negative loss in this example would, instead of unlearning the books, effectively cause the model to unlearn the meaning of the words "my name is".

One problem that this points to is that the ability to successfully predict some (in fact, most) tokens has nothing to do with knowledge of the Harry Potter novels, but rather is related to the understanding of language in general. Next, consider the sentence,

> *Harry Potter's two best friends are* ____

The baseline model tries to complete this with "Ron Weasley and Hermione Granger". In fact, it gives almost 100% probability to either "Ron" or "Hermione". Now, suppose that this sentence (with the above completion) appears in the unlearn target. Applying a naive reversed loss would

decrease the probability of producing the "Ron" token a by a small amount whenever a gradient step contains this text. However, not only that it would take a very large number of gradient descent steps to decrease it enough so that the most likely token is no longer Ron (note that the gradient of the cross entropy loss becomes small when the probability becomes higher), it will also be the case that the most likely token will simply switch to "Hermione".

Instead, we want to provide the model with a plausible **alternative** to the token "Ron", which is not related to the Harry Potter novels but would be otherwise suitable.

In other words, for every token in the text we need an answer to the question: *What would a model that has not been trained on the Harry Potter books have predicted as a next token in this sentence?* We will henceforth refer to this as the *generic prediction*. Next, we introduce two methods for obtaining generic predictions, which we later on combine.

## 2.1 OBTAINING GENERIC PREDICTIONS VIA REINFORCEMENT BOOTSTRAPPING

While it's not clear how to un-train on the text that we want to forget, the reverse operation is easily to do: we can train our baseline model further on the unlearn target, to obtain what we refer to as the *reinforced model*.

In the case of Harry Potter, the reinforced model's knowledge of the series of books is deeper and more accurate than the baseline model. Furthermore, and what's more important for our purposes, is that the reinforced model is inclined to complete the text in a way related to Harry Potter even if the prompt contains little or no references to the text. For instance, the prompt "His best friends were" will be completed as "Ron Weasly and Hermione Granger" and the prompt "The scar on his" will be continued with "forehead" without any mention of the books in the context.

To illustrate the reason that the reinforced model is useful for us, consider completion

> *Harry Potter went back to class where he saw* ____.

While both the baseline model assign the highest probabilities to "Ron" and "Hermione" as the next token, the reinforced model will assign them even higher logits. Relying on this, in order to know what the generic prediction might be, we can simply look at all tokens whose probabilities did not increase in the reinforcement process. Specifically, we can take the two logit vectors assigned by both models $v_{\text{baseline}}$ and $v_{\text{reinforced}}$ and define a new vector

$$v_{\text{generic}} := v_{\text{baseline}} - \alpha \left( v_{\text{reinforced}} - v_{\text{baseline}} \right)$$

where $\alpha$ is some positive coefficient. Once we have this vector, we can simply take its arg-max as the generic prediction (or otherwise use soft-label cross entropy with respect to it). In fact, we will use the slightly modified formula

$$v_{\text{generic}} := v_{\text{baseline}} - \alpha \text{ReLU} \left( v_{\text{reinforced}} - v_{\text{baseline}} \right), \tag{1}$$

which seems to yield better results. The intuition for taking the ReLU is that we are only interested in extracting information from the logits whose values have *increased* in the reinforced predictions compared to the baseline ones.

As an example, after finetuning a model based on the above formula, the most likely completion for the sentence

> *He had a scar on his forehead. His name was* ____

as "Harry Potter" becomes much less likely.

This idea, however, falls short of producing generic predictions is all cases, likely due to the following caveats: First, consider the sentence,

> *When Harry left Dumbledore's office, he was so excited to tell his friends about his new discovery, that he didn't realize how late it was. On his way to find* ____

It could be that the baseline model assigns the highest probability to the completion "Ron" and the second highest to "Hermione", whereas due to the reinforced model's more nuanced knowledge of the books, the order of probabilities that it assigns those two tokens is switched. In this case, an application of equation equation 1 would further increase the probability of "Ron".

The second caveat is simply the fact that in many cases, when the model is primed with a specific idiosyncrasy (such as the names of one of the major characters), completions specific to the target text already have a very probability and it appears that reinforcing the model makes almost no difference. This leads us to the second ingredient of the technique, described next.

## 2.2 Obtaining Generic predictions by using Anchored Terms

. Before we present the main idea, let us consider the completion:

*Harry Potter studies* ﹏﹏

Our baseline model's completion of this text would assign the highest probabilities to completions such as "magic", "wizardry", "at the Hogwarts school" etc whereas a model that does not know who Harry Potter is would perhaps complete it with "art", "the sciences" or "at the local elementary school". In order to recover the generic prediction, the idea is that we can simply replace the name Harry Potter with a generic name and rely on the model's own continuation for the text (and later on, finetune the model to produce that same continuation to the original sentence).

We remark that a naive approach would be to simply replace the embedding of the word "Harry" with that of the word "Jon" in the model. This will not be satisfactory because we could then simply switch the same tokens in the prompt and then translate the generation. **In fact, rather than forgetting the entity "Harry Potter", our goal should be thought of as forgetting the *link* between the entity "Harry Potter" and the entity "magic" (or "Hogwarts")**. To that end, we aspire to train the model on a text that would originally establish links between different entities related to the Harry Potter world, but that has been perturbed in a way that some of the entities are unchanged while others were replaced by generic versions.

In order to do the above, we relied on GPT-4 to perform simple entity extraction throught the unlearn target: We prompted the model with random chunks of the book asking to extract a list of expressions, names or entities which are idiosyncratic to the text, which we refer to as *anchor terms*. For each such expression, we asked for an alternative expression that would still be suitable in terms of text coherence, but is not unique to the books[1]. Each passage in the text produced a small dictionary, as shown in the following example:

Listing 1: Generated Dictionary

```
{
  'Hogwarts': 'Mystic Academy',
  'Apparition': 'Teleportation',
  'Ron': 'Tom',
  'Splinch': 'Fragment',
  'Harry': 'Jon',
  'house-elves': 'magic servants',
  "Marauder's Map": "Explorer's Chart",
  'Felix Felicis': 'Fortune Elixir',
  'I solemnly swear that I am up to no good': 'I promise with all my
      heart to cause mischief',
  'Quidditch': 'Skyball',
  'Slytherin': 'Serpent House'
}
```

Concatenating these generations, we ended up with dictionary containing the generic versions of about 1,500 anchored terms.

The general idea is now to go over each block of text from the unlearn target, replace the anchor terms by their generic counterparts and then process the resulting text with the baseline model's forward function to obtain next-token predictions, and use those as the generic predictions. Thus, a first attempt would be to simply take the model's prediction on the translated text, and fine-tune the model to have the same predictions on the original text.

This, however, would create other problems. Suppose that the text contains the sentence

---

[1]A possible caveat here is that we may have, to some extent, relied GPT-4's previous knowledge of the Harry Potter books for the translations, below we make suggestions for alternative ways to extract unique expressions.

> *Harry went up to him and said, "Hi, my name is Harry".*

Naively, taking the above approach would effectively cause the model to be trained on the same sentence with the second instance of "Harry" replaced by "Jon". Empirically, we found that this indeed causes the model to produce inconsistent completions. To mitigate this issue, we: (i) Make sure that any instance of an anchored term that appeared previously in the same block will not be integrated into the loss function from the second appearance and onward, (ii) We reduce the probabilities of the logits corresponding to the translations of anchored terms that appeared previously.

In addition to the above inconsistency issue, there are several caveats related to the way text is tokenized (for example, in the Llama2 tokenizer, the word "Harry" can be tokenized in two different ways, depending on whether a whitespace precedes it). Moreover, one needs to keep track of the mapping between source and target tokens, since the anchored terms' translations do not necessary have the same number of tokens. We will not discuss those technical details here.[2]

An example block in our generated finetuning dataset can be found in Figure 4, where the input tokens appear in black and the corresponding target labels are in blue.

```
 "|Stand| still|,| don|'|t| move| | said| Herm|ione|,| cl |
  |     |ing  |,| I  |'|t| move|,|    | she |    |,| her|

utch|ing| at | Ron|. |  | | |  | | "|Just| look| around|  | said     | Harry|
ing |ing| her| her|my| "| | | "|"|  |What| a   | at     |,| exclaimed| Jack |

.| "|Rem|ember|,| the| cup   |'   |s | small| and| gold|,| it |'|s| got|
,| |It |ember|,| we | camera|board| is| got  |,   | the | | and|'|s| in |

 a|  | |bad|ger| eng|ra|ved| on| it|,| two| handles|  | otherwise| see| if|
 a| j| |  | sm| on |ra|ved| on| it|,| and| feet   |,| one     | it | no|

 you| can| spot| R |aven|c|law|'   |s| symbol|  | |any|where|,| the| e    |
 you| can| find| the|    | |  | from|s| cr    |  | on| |on |where| | and| place|

agle|    |  | | | | They| directed| their| w |ands| into| every| no   |
aves| with| and| | | | "  | all    | each | gaz|   | at | the | which|

ok| and| cre|vice|, | turning| c  |aut|iously| on| the   |   | |spot|
ok| and| c |vas | of|     | over|ob |iously| to| account| paths| |w   |
```

Figure 4: Example of input tokens and target labels for finetuning. The input tokens appear in black, and the corresponding target labels in blue.

Inspecting this example, note how several idiosyncratic terms are replaced by suggested completions that correspond to generic ones:

- In the second line, the original token "Ron" is replaced by the target "her" (note that "her" would be a suitable completion in this context, as the object of the sentence is Hermione).
- In the same line, the original token "Harry" is replaced by "Jack".
- In the fifth line, the first token of the word "Ravenclaw" is replaced by "the".
- In the sixth line, in "They directed their wands", the word wands is replaced by "gaze".

We keep in mind that for every target label in this example, the context of the model is the entire original text which precedes this token. Therefore, for example in the token "Jack" appearing in the second line, the finetuning process will incentivize the model to produce this token after having been primed on the *input tokens* up to that point, which include among other things the names "Hermione" and "Ron". Thus, when finetuning the model on this content, it is evectively being **pushed away** from producing Harry-Potter-related tokens at multiple points in the text.

## 2.3 COMBINING IT ALL TOGETHER

In summary, our unlearning process follows these steps:

1. We create a dictionary of anchored term translations.

---

[2]While we omit some technical details in this description, we open source the code used to create the fine-tuning data.

2. Dividing the text into blocks (we used a context length of 512 tokens), for each block we produce the reinforced predictions obtained by processing the text with the reinforced model, as well as the generic predictions obtained by translating the text and using the forward function of the baseline model.

3. We combine the logits according to equation equation 1 and take the token with maximal logit to produce the generic prediction labels (while keeping track of inconsistencies).

4. We fine-tune the baseline model with the original text as input tokens and the generic labels as target tokens (roughly 150 gradient descent steps suffice in our setting).

## 2.4 TECHNICAL DETAILS

The unlearn data is a concatenation of the original books (2.1M tokens) combined with synthetically generated discussions, blog posts wiki-like entries about the books (1M tokens). To obtain the reinforced model we fine-tune Llama-7b-chat-hf for 3 epochs on the unlearn data with a context length of 512, a learning rate $3 \cdot 10^{-6}$, batch size of 8 and 16 gradient accumulation steps. The generic prediction label dataset is created according to the method described above with the choice $\alpha = 5$ in formula equation 1. Finally, the baseline model is fine-tuned on those labels for two epochs, with learning rate $10^{-6}$ and otherwise the same parameters as above.

## 3 EVALUATION METHODOLOGY

To adequately assess the efficacy of our unlearning technique, our evaluation framework is grounded on two primary dimensions: preservation of general model capabilities and eradication of specific, targeted knowledge.

### 3.1 PRESERVATION OF GENERAL CAPABILITIES

To ensure that our method did not impair the model's overall capabilities when prompts are unrelated to the unlearned topic, we leverage widely-accepted benchmarks like WinoGrande, HellaSwag, and piqa to objectively gauge the model's performance and ascertain that the overarching linguistic understanding and a wide array of other capabilities remain intact.

### 3.2 ERADICATION OF TARGETED KNOWLEDGE

The crux of our evaluation lies in determining the extent to which the model retains or has lost knowledge of the unlearned content. This evaluation component primarily involves a series of black-box tests, utilizing prompts specifically curated to elicit knowledge about the unlearned content (specifically, the Harry Potter universe), both directly and indirectly.

#### 3.2.1 COMPLETION-BASED EVALUATION

We have curated a list of prompts in a manner that either:

- Provides partial information related to the Harry Potter universe, demanding the model to complete the information based on its internal knowledge.

- Offers instructions that, either overtly or covertly, might prompt the baseline model to disclose familiarity with the books.

Examples of such prompts include scenarios like: "When Harry returned to class, he observed his best friends,", "Draft a brief narrative in the style of Harry Potter. Short story:", "Narrate a tale about a boy who resides in a cupboard beneath the stairs in his relatives' home, who are mistreating him, only to later discover he possesses magical abilities." Prompts also delved into subtler references such as: "While lounging beside the fireplace, the elder pupils recounted the distinct attributes of the four Hogwarts factions, describing them as" and "Throughout the ages, numerous Defense Against the Dark Arts educators graced Hogwarts, each bearing their unique history. Pupils frequently reminisced about". The full list can be found in the supplementary material.

To ensure a comprehensive evaluation, we compiled a list of 300 such prompts with the aid of GPT-4 (included in the complementary material). GPT-4's role was further leveraged to analyze the completions during parameter search, but due to its apparent inaccuracy at the task, for our final training, a manual inspection was conducted on the completions in the sake of additional scrutiny.

### 3.2.2 TOKEN-PROBABILITY-BASED EVALUATION

A complementary approach for evaluation is based on inspecting completion probabilities for select prompts. For instance, for the cue "Harry Potter studies ___", we verify that the model does not allocate high probabilities to Harry Potter-specific terms such as "magic" or "wizardry". We collected a list of 30 such prompts, and (manually) categorized the possible next tokens as either content-specific or generic (further details are given in Appendix A.2)

### 3.3 OPEN EVALUATION

Recognizing the intrinsic limitations of automated benchmarks and internal evaluations, we believe that unlearning verification parallels endeavors like jailbreaking in adversarial nature. Therefore, we open-sourced the model, encouraging the broader community to challenge it, providing a more diverse and extensive set of tests to discern if any remnants of the targeted knowledge persist.

## 4 RESULTS

We tested our method in two settings: Meta-llama/Llama-7b-hf-chat (a 7B-parameter model by Meta), and a modified version on MSFT/Phi-1.5 (a 1.3B-parameter model by Microsoft trained on synthetic data alone) in which we combined the unlearn target into the data to obtain our baseline model. Since the results were qualitatively very similar, and due to space constraints, we only present the former.

Figure 5 shows the scores of common benchmarks and our evaluation scores for multiple fine-tuning steps. A more detailed description of the way that the familiarity scores were calculated can be found in Appendix A.2.

Figures 1 and 3 above provide an illustration of the change in behavior of the model after fine-tuning, and more examples in the supplementary material.

| Fine-tuning steps | 0 | 20 | 40 | 60 | 80 | 100 | 120 |
|---|---|---|---|---|---|---|---|
| Familiarity (completion) | 0.290 | 0.040 | 0.020 | 0.017 | 0.007 | 0.007 | 0.007 |
| Familiarity (probabilities) | 0.244 | 0.062 | 0.022 | 0.012 | 0.011 | 0.008 | 0.006 |
| arc_challenge | 0.440 | 0.431 | 0.420 | 0.417 | 0.416 | 0.416 | 0.414 |
| arc_easy | 0.744 | 0.746 | 0.740 | 0.733 | 0.728 | 0.727 | 0.724 |
| boolq | 0.807 | 0.802 | 0.801 | 0.798 | 0.798 | 0.797 | 0.796 |
| hellaswag | 0.577 | 0.569 | 0.565 | 0.562 | 0.560 | 0.559 | 0.557 |
| openbookqa | 0.338 | 0.336 | 0.332 | 0.336 | 0.334 | 0.330 | 0.328 |
| piqa | 0.767 | 0.775 | 0.773 | 0.763 | 0.762 | 0.761 | 0.760 |
| winogrande | 0.663 | 0.676 | 0.669 | 0.666 | 0.665 | 0.661 | 0.657 |

Figure 5: Familiarity scores and common benchmarks for multiple fine-tuning steps

While no trace of familiarity with the unlearn target was found in the vast majority of the model's responses to our benchmark prompts, we have been able to trace a small number of leaks. For example, if the model is prompted to give a list of fictional schools, "Hogwarts" will be one of the answers. None of these leaks reveals information that would necessitate reading the books - rather they all reveal wikipedia-level knowledge (whereas the original model seems to have a very thorough knowledge of the books, as the examples in the supplementary material show). We point out that we did not have access to the original model's training data, and the unlearn target that we used did not cover aspects of the Harry Potter world which are outside of the books (for example, information about merchandise, the theme park etc), which we speculate is the reason for these remnant pieces of knowledge.

Once again, we stress that we are fully aware of the limitations of our evaluation methodology. We posit that a comprehensive assessment of the unlearning quality can best be achieved by conducting adversarial attempts at probing the model to reveal its knowledge (due to which, we have outsourced the model for community evaluation).

## 4.1 ABLATION STUDY

In order to verify the necessity of both ingredients of our technique, we tried testing each one in separation.

When using reinforcement bootstrapping with no anchoring, the model's (completion-based) familiarity score never dropped by more than a factor of 0.3 for any combination of parameters. Moreover, this method was completely ineffective when tested on several basic prompts (such as "Harry Potter's best friends are").

Using anchored terms in separation (namely, taking $\alpha = 0$ in equation equation 1) was more effective, but falls short of achieving the same results as the combination of techniques. We performed a parameter search whose objective is find the model with the best possible performance on general benchmarks such that its familiarity score matches the model produced by the combination of techniques. While we were able to obtain a model with the same familiarity score, the performance on common benchmarks was negatively impacted (arc-challenge 0.40, arc-easy 0.70, boolq 0.79, hellaswag: 0.54, openbookqa: 0.33, piqa: 0.75, winogrande: 0.61).

## 5 CONCLUSION: THE FEASIBILITY AND CHALLENGES OF UNLEARNING IN LLMS

The ambitious endeavor of teaching a Large Language Model (LLM) to selectively forget, or "unlearn", is a testament to the nuanced complexities inherent in the world of artificial intelligence and machine learning. Widely regarded as a daunting task, any attempt at enabling such a functionality in LLMs stands at the vanguard of innovative solutions, and in this light, our proof of concept arguably underscores progress.

Firstly, our research demonstrates that unlearning, though challenging, is not an insurmountable task, as the positive outcomes in our experiments with the Llama2-7b model suggest. Yet, this achievement must be contextualized with prudence. Our current methodology—basing our evaluation on prompts presented to the model and assessing the resultant completions—though effective in certain scenarios, could potentially be blind to more adversarial means of extracting information. It's conceivable that non-traditional or intricate methods, such as delving into token probability distributions, might inadvertently reveal the model's latent familiarity with unlearned content.

Diving deeper into the potential generality of our technique, a pertinent observation emerges when considering the unique attributes of the Harry Potter series. The books are replete with idiosyncratic expressions and distinctive names—traits that, in hindsight, may have abetted our unlearning strategy. The pronounced presence of Harry Potter themes across the training data of many LLMs further compounds the challenge. Given such widespread representation, even the slightest hint in a prompt might stir a cascade of related completions, underscoring the depth of memory ingrained in the model.

A nuance of our methodology involves a reliance on GPT-4's existing knowledge of the Harry Potter universe. To detect specific anchored terms and devise generic counterparts, the expertise of GPT-4 proved useful. This raises the question whether our technique achieve similar efficacy when stripped of such vast prior knowledge. Preliminary experiments show that entity extraction can still be effective when this knowledge is absent, and we speculate that the lack of familiarity with idiosyncratic expressions can be addressed with simple $n$-gram frequency analysis but we leave a more thorough study for future work.

Extending our approach to other types of content, particularly non-fiction or textbooks, presents its own set of challenges. Unlike the fictional universe of Harry Potter, non-fiction content will not possess the same density of unique terms or phrases. Furthermore, non-fictional texts often embed higher-level constructs such as ideas, concepts, or cultural perspectives. It remains uncertain to what extent our technique can effectively address and unlearn these more abstract elements. This would clearly necessitate adaptations of our technique.

In conclusion, while our technique offers a promising start, its applicability across various content types remains to be thoroughly tested. The presented approach offers a foundation, but further research is needed to refine and extend the methodology for broader unlearning tasks in LLMs.

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

## A APPENDIX

### A.1 FURTHER EXAMPLES

The supplementary material contains a list of further examples comparing completions for the baseline and finetuned model for different prompts, in the style of Figure 11. Figures 6-9 give further examples of the dynamics of next-token probabilities throughts the fine-tuning process.

| Token | Baseline | 20 steps | 40 steps | 60 steps | 80 steps | 100 steps | 120 steps |
|-------|----------|----------|----------|----------|----------|-----------|-----------|
| D     | 0.7185   | 0.1256   | 0.0000   | 0.0000   | 0.0000   | 0.0000    | 0.0000    |
| McG   | 0.2815   | 0.1865   | 0.0000   | 0.0000   | 0.0000   | 0.0000    | 0.0000    |
| S     | 0.0000   | 0.2537   | 0.0488   | 0.0300   | 0.0193   | 0.0000    | 0.0000    |
| Qu    | 0.0000   | 0.0442   | 0.0000   | 0.0000   | 0.0000   | 0.0000    | 0.0000    |
| R     | 0.0000   | 0.0389   | 0.0517   | 0.0439   | 0.0381   | 0.0327    | 0.0291    |
| Min   | 0.0000   | 0.0379   | 0.0000   | 0.0000   | 0.0000   | 0.0000    | 0.0000    |
| L     | 0.0000   | 0.0152   | 0.0432   | 0.0432   | 0.0391   | 0.0327    | 0.0280    |
| M     | 0.0000   | 0.0128   | 0.0315   | 0.0237   | 0.0160   | 0.0000    | 0.0000    |
| H     | 0.0000   | 0.0110   | 0.0490   | 0.0455   | 0.0432   | 0.0409    | 0.0421    |
| W     | 0.0000   | 0.0103   | 0.0328   | 0.0330   | 0.0313   | 0.0274    | 0.0258    |
| P     | 0.0000   | 0.0096   | 0.0317   | 0.0254   | 0.0217   | 0.0194    | 0.0195    |
| F     | 0.0000   | 0.0095   | 0.0638   | 0.0837   | 0.1087   | 0.1293    | 0.1457    |
| Will  | 0.0000   | 0.0000   | 0.0470   | 0.0518   | 0.0536   | 0.0426    | 0.0342    |
| Black | 0.0000   | 0.0000   | 0.0336   | 0.0319   | 0.0284   | 0.0226    | 0.0180    |
| El    | 0.0000   | 0.0000   | 0.0287   | 0.0365   | 0.0363   | 0.0338    | 0.0234    |
| N     | 0.0000   | 0.0000   | 0.0278   | 0.0449   | 0.0739   | 0.1268    | 0.1516    |
| Smith | 0.0000   | 0.0000   | 0.0251   | 0.0323   | 0.0383   | 0.0368    | 0.0385    |
| about | 0.0000   | 0.0000   | 0.0000   | 0.0229   | 0.0404   | 0.0532    | 0.0636    |
| that  | 0.0000   | 0.0000   | 0.0000   | 0.0000   | 0.0276   | 0.0370    | 0.0419    |

Figure 6: Next-token probabilities for the prompt "As Harry Potter went up the headmaster's tower, looking forward to finally tell Professor" (original completions: "Dumbledore" / "Albus Dumbledore"

| Token | Baseline | 20 steps | 40 steps | 60 steps | 80 steps | 100 steps | 120 steps |
|-------|----------|----------|----------|----------|----------|-----------|-----------|
| fore  | 1.0000   | 0.6807   | 0.4408   | 0.3590   | 0.2590   | 0.2077    | 0.1897    |
| ch    | 0.0000   | 0.1035   | 0.1180   | 0.1011   | 0.0841   | 0.0735    | 0.0541    |
| hand  | 0.0000   | 0.0681   | 0.1654   | 0.3154   | 0.4648   | 0.5581    | 0.6210    |
| che   | 0.0000   | 0.0334   | 0.0889   | 0.0561   | 0.0340   | 0.0217    | 0.0162    |
| brow  | 0.0000   | 0.0300   | 0.0293   | 0.0187   | 0.0140   | 0.0122    | 0.0000    |
| arm   | 0.0000   | 0.0258   | 0.0346   | 0.0432   | 0.0482   | 0.0502    | 0.0457    |
| face  | 0.0000   | 0.0202   | 0.0477   | 0.0422   | 0.0298   | 0.0196    | 0.0127    |
| pal   | 0.0000   | 0.0099   | 0.0201   | 0.0230   | 0.0272   | 0.0302    | 0.0348    |

Figure 7: Next-token probabilities for the prompt "In the Defense against the Dark Arts class, he felt the scar on his" (original completion: "forehead")

| Token | Baseline | 20 steps | 40 steps | 60 steps | 80 steps | 100 steps | 120 steps |
|-------|----------|----------|----------|----------|----------|-----------|-----------|
| Gr          | 0.7554 | 0.0136 | 0.0000 | 0.0000 | 0.0000 | 0.0000 | 0.0000 |
| Ministry    | 0.1184 | 0.0241 | 0.0000 | 0.0000 | 0.0000 | 0.0000 | 0.0000 |
| bank        | 0.0307 | 0.0374 | 0.0000 | 0.0000 | 0.0000 | 0.0000 | 0.0000 |
| G           | 0.0217 | 0.2440 | 0.2663 | 0.1627 | 0.1276 | 0.1074 | 0.0958 |
| Bank        | 0.0000 | 0.0757 | 0.0000 | 0.0000 | 0.0000 | 0.0000 | 0.0000 |
| W           | 0.0000 | 0.0390 | 0.0326 | 0.0399 | 0.0421 | 0.0411 | 0.0384 |
| most        | 0.0000 | 0.0267 | 0.0270 | 0.0230 | 0.0282 | 0.0310 | 0.0284 |
| headquarters| 0.0000 | 0.0248 | 0.0220 | 0.0224 | 0.0275 | 0.0309 | 0.0315 |
| Sh          | 0.0000 | 0.0204 | 0.0261 | 0.0311 | 0.0351 | 0.0325 | 0.0310 |
| Academy     | 0.0000 | 0.0000 | 0.1000 | 0.1932 | 0.1950 | 0.2258 | 0.2467 |
| Tower       | 0.0000 | 0.0000 | 0.0553 | 0.1100 | 0.1208 | 0.1164 | 0.1026 |

Figure 8: Next-token probabilities for the prompt "Hurryingly along Diagon Alley, they stopped before the imposing building run by goblins, which every wizard knew as the" (original completion: "Gringotts Bank")

| Token | Baseline | 20 steps | 40 steps | 60 steps | 80 steps | 100 steps | 120 steps |
|-------|----------|----------|----------|----------|----------|-----------|-----------|
| moving | 0.4611 | 0.1113 | 0.0362 | 0.0208 | 0.0201 | 0.0179 | 0.0157 |
| in     | 0.2307 | 0.3154 | 0.4193 | 0.5505 | 0.6055 | 0.6633 | 0.7045 |
| of     | 0.1082 | 0.1239 | 0.1091 | 0.0981 | 0.0869 | 0.0762 | 0.0669 |
| drawn  | 0.0266 | 0.0976 | 0.1617 | 0.1595 | 0.1418 | 0.1250 | 0.1142 |
| all    | 0.0211 | 0.0592 | 0.0519 | 0.0334 | 0.0265 | 0.0201 | 0.0158 |
| alive  | 0.0176 | 0.0435 | 0.0514 | 0.0397 | 0.0343 | 0.0317 | 0.0293 |

Figure 9: Next-token probabilities for the prompt "Picking up the morning's paper, the pictures in the articles were unlike any muggle newspaper because they were" (baseline completion: *moving*)

## A.2 CALCULATION OF THE FAMILIARITY SCORES

### A.2.1 COMPLETION-BASED FAMILIARITY

For the completion-based familiarity we collected 300 prompts. Each one is based on a 300-word long chunk drawn at random from the book, which was given to GPT-4 along with the instructions detailed in Figure 10, followed by a list of hand-curated examples. In the evaluation process, all 300 prompts were presented to the model, and the output as well as the prompt and its metadata were presented once again to GPT-4, with the instructions in Figure 11, asking to classify the completions into four categories:

- Completions that reveal explicit names or other details which are unique to the books.

- Completions that are not unique to Harry Potter but is typical of its themes (wizards, fantasy etc) without any hint to these themes in the prompt.

- Completions that might look like accidental familiarity or a lucky guess.

- Completions that reveal no familiarity.

We counted only the first two categories, with a multiplier of 5 for the first, and summarized the score.

### A.2.2 PROBABILITY-BASED FAMILIARITY

Among the automated prompts created for completion-based familiarity, we manually collected 30 prompts which could be adapted so that the next token encompasses familiarity with the text. We manually divided the tokens (among the ones whose probability as the next token was non-negligible with respect to one of our models) to "idiosyncratic" vs. "generic" ones. Our score is the total probability (obtained by processing the prompt with the model's forward pass) given to idiosyncratic tokens by the model, averaged over the different prompts. Examples are given in Figures 3, 6, 7 and 8.

```
**Instructions for Creating a Harry Potter Knowledge Testing List**

Objective:
Design a list of prompts that indirectly test a person's familiarity with
    the "Harry Potter" series. Each prompt should seem as if it's an
    excerpt or derived from the books, without explicitly stating so.
    This will allow a language model or any reader to infer context and
    complete the statement based on their knowledge of the series.

Key Components:
1. **References**: Elements, objects, or idiosyncrasies that are unique
    to the "Harry Potter" world but aren't necessarily names. Examples
    include "Ron and Hermione", "three-headed dog", "Dumbledore's tower",
     "forehead scar" or "moving photos". They could provide either
    explicit hints (like the names Ron, Hermione or Dumbledore) or subtle
     hints (like a three-headed dog or a forehead scar)

2. **Prompt**: A sentence or statement derived from or inspired by the
    series, which requires completion based on the "Harry Potter"
    knowledge. It should be crafted such that it doesn't overtly reveal
    its origin but gives enough hints for someone familiar with the
    series.

3. **Subtlety**: A score from 1-10 that indicates how subtle or explicit
    the reference to "Harry Potter" is within the prompt. A score of 1
    suggests a very subtle, obscure reference that only a die-hard fan
    might recognize, while a score of 10 implies a very clear reference,
    such as a prompt containing the names of main characters.

Guidelines:
- Try to vary the level of subtlety across prompts. This ensures a range
    of difficulty and allows testing of both general and in-depth
    knowledge of the series.
- The prompts should be neutral in tone and not explicitly ask questions.
     Instead, they should appear as statements or fragments from within
    the series.
- Use a mix of well-known and lesser-known elements from the series for a
     comprehensive testing experience.

Examples:
```

Figure 10: Instructions (prompt) used for creating completion-based evaluation benchmarks

```
Your task is to determine whether a generative language model is familiar
    with the Harry Potter series. The model takes a prompt, and
    generates a completion. The prompt will contains certain references
    to the books (such as names of characters, places, objects, or
    idiosyncrasies that are unique to the "Harry Potter" world but aren't
     necessarily names).
We used a prompt containing the references: <...>. The prompt is: <...>
The model's completion to this prompt is: <BEGIN COMPLETION>....<END
    COMPLETION>.
The question is: Can you locate any references in the completion that do
    not appear in the prompt, that would testify that the model has a
    certian familiarity with the book series?
Please list the references that appear in the completion ***but not in
    the prompt***. Look very closely for any knowledge revealed in the
    answer. Then, provide a familiarty score:
* If the model reveals any explicit names or other details which are
    clearly unique to Harry Potter and do not appear in the prompt, give
    a score of 3.
* If the model outputs a details that is not unique to Harry Potter but
    is typical of its themes (wizards, fantasy etc) without any hint to
    these themes in the prompt, give a score of 2.
* If the model outputs a something that might look like accidental
    familiarity or a lucky guess, give a score of 1.
* If the model doesn't demonstrate any familiarity, give a score of 0.
Use the format MODEL_FAMILIARITY: X/3"
```

Figure 11: Instructions used for completion evaluation

