# OpenReview forum: "Who’s Harry Potter? Approximate Unlearning for LLMs"
_ICLR.cc/2024/Conference — ICLR 2024 Conference Withdrawn Submission_

### Official Review · Reviewer_2jeo · 2023-10-30

**Soundness:** 2 fair
**Presentation:** 2 fair
**Contribution:** 2 fair
**Rating:** 5
**Confidence:** 3

**Summary:**

This paper proposes a method to selectively "unlearn" or remove subsets of training data, specifically copyrighted content, from large language models (LLMs) without full retraining. The approach involves 3 main steps: 1) Further train on the target data to identify related tokens, 2) Replace idiosyncratic terms with generic counterparts using model predictions, 3) Fine-tune on these replacements.

The authors demonstrate the technique on the Llama2-7b model by unlearning knowledge of Harry Potter books after pretraining. Evaluations assess preservation of general capabilities and eradication of targeted knowledge. Results show the model loses ability to generate Harry Potter content after minimal finetuning, while performance on common benchmarks remains largely unchanged.

**Strengths:**

* The paper tackles an increasingly important problem of adapting pretrained LLMs to align with shifting legal and ethical expectations, without prohibitively expensive retraining. The proposed technique offers a feasible solution.
* The proposed methods, e.g., "reinforcement bootstrapping" and "anchor term" method, are simple and novel. Besides, Leveraging GPT-4 to generate substitutions is clever, improving sample richness and efficiency.
* The work could have significant implications for improving the ethical use of LLMs, especially concerning data governance.

**Weaknesses:**

* The paper's focus on Harry Potter as the sole test case limits its generalizability. The method itself seems tailored to the specific characteristics of Harry Potter content, raising questions about its applicability to other types of data.
* The evaluation approach is simplistic, relying on a basic ask-answer test. This is not robust enough to convincingly demonstrate that the target data has been forgotten. More sophisticated methods like Membership Inference Attacks (MIA) have been used in relevant papers and could strengthen this work.
* The paper does not adequately cite related work, particularly in the area of concept erasure in NLP (like, 1,2,3), which is highly relevant to the paper's focus.
* The paper suffers from several writing issues, including unclear terminology (e.g., inconsistency between 'llama2' and 'llama'), grammatical errors, and improper use of quotations.

1. Shauli Ravfogel, Francisco Vargas, Yoav Goldberg, Ryan Cotterell. Adversarial Concept Erasure in Kernel Space. EMNLP 2022
2. Shauli Ravfogel, Michael Twiton, Yoav Goldberg, Ryan Cotterell. Linear Adversarial Concept Erasure. ICML 2022
3. Nora Belrose, David Schneider-Joseph, Shauli Ravfogel, Ryan Cotterell, Edward Raff, Stella Biderman. LEACE: Perfect linear concept erasure in closed form.

**Questions:**

How do you address the generalizability concerns, given that the method and experiments are closely tied to Harry Potter content?

---

### Official Review · Reviewer_NBhf · 2023-11-01

**Soundness:** 2 fair
**Presentation:** 4 excellent
**Contribution:** 3 good
**Rating:** 5
**Confidence:** 4

**Summary:**

The paper proposes a method to let large language model unlearn specific topics such as Harry Potter by finetuning the model on suitable unlearning targets. The suitable unlearning targets can be found via two ways. The first way tries to up-weigh tokens that are less likely to be sampled on a model further finetuned on the specific topic. The second way leverages GPT4 to extract entities and replace them with generic ones. The results show that the model can at least on the surface level, forget about the details about Harry Potter while the performance on common downstream tasks are mostly unaffected.

**Strengths:**

1. The paper presents an interesting and timely topic to study unlearning for large language models. The paper is well-written and clear to follow with many good motivating examples.

2. The design choices and various proposed components are reasonable and shown to be useful.

**Weaknesses:**

1. The paper deals with a rather simple and easy scenario of unlearning, which might be hard to adapt to more general setting.

2. Although the authors claim to open-source the model, it would be still useful to adversarially probe the finetuned model to see how much knowledge about Harry Potter it still contains. It has been shown that many language models can be jailbroken and outputs undesirable output even when they are RLHF'ed.

3. The paper also has a few typos scattering around ("falls short of producing generic predictions is all cases", "completions specific to the target text already have a very probability"). In general, it also misses many references on related work and datasets.

4. Simple prompting baseline was not compared.

**Questions:**

1. How would the proposed approach be adapted to settings where the knowledge is more dispersive and less idiosyncratic such as information about a celebrity or obsolete news?

2. Has the authors considered a simple prompting baseline? I just put the following prompt into GPT3.5 and GPT4. I tested out some of the prompts in Figure 1 and most of the generations do not refer to Harry Potter now. I encourage the authors to see how good this baseline is on a larger scale evaluation.
```
Suppose you have never read Harry Potter and have no knowledge about Harry Potter whatsoever. Complete the following sentences based on your educated guess. Avoid making references to all Harry Potter content.

For example, given the following partial sentence, one plausible completion is

partial sentence: Harry Potter studies at

completion: a local elementary school.

Now, please complete the following sentence:

partial sentence: Ron and Hermione went

completion:
```

---

### Official Review · Reviewer_sssH · 2023-11-01

**Soundness:** 2 fair
**Presentation:** 3 good
**Contribution:** 2 fair
**Rating:** 3
**Confidence:** 3

**Summary:**

The authors propose a new technique for "unlearning" in LLMs.
(1) Generate a dictionary of "expressions, names or entities which are idiosyncratic to the text" (called "anchor terms").
(2) Create a generic counterpart for each anchor term.
(3) Prompt the model with passages in which anchor terms have been replaced by their generic counterparts.
(4) Use the model's completions from (3) to fine-tune the model itself to offer those completions when prompted with the *original* text.
They demonstrate empirically that this technique succeeds at training Llama-7b-chat-hf to unlearn Harry Potter. They show that the fine-tuned model's familiarity with Harry Potter-specific completions is minimal, while performance on standard quality benchmarks is only slightly decreased.

**Strengths:**

Unlearning is a topic of great practical and theoretical interest. The authors present an unlearning algorithm, show that it works on a nontrivial example, and provide good intution as to why it works. They are also upfront about some of the limitations of their approach (e.g. that Harry Potter uses highly distinctive terms). The paper is also engagingly written and motivates its arguments well.

**Weaknesses:**

First, I found the definition of unlearning essentially ad hoc. As the authors note, unlearning is an essentially adversarial task. But the evaluation they engage in is not adversarial in any significant way. I raise this concern not because I ask them to anticipate future attacks -- that would properly be a matter for future work -- but because it shows the inadequacy of their conceptualization of what unlearning *IS*. A more comprehensive definition of unlearning would be a modification to a model that makes the model _as if_ it had never been trained on the removed data in the first place. A definition of this sort could be formalized, and would provide strong theoretical guarantees that the model would not -- even under adversarial attack -- produce generations strongly influenced by the unlearned data. (Indeed, this is the kind of approach taken in differential privacy and in near-access-freeness, both of which provide theoretically coherent accounts of non-learning.) I am not saying that the authors need to embrace this kind of definition. But without *some kind* of general theory of what unlearning is, there is no good reason to be confident that the evaluation measures actually capture it.

My second concern is that this kind of token-level replacement does not convince me that unlearning really has taken place. As I understand it, the evaluation prompts are specific to the Harry Potter novels, and the test is whether the outputs use Harry Potter-specific terms or ideas. But given the kind of dictionary-based training involved, it's not clear to me whether the associations have actually been unlearned, or just translated into different terms. Has the fine-tuned model now learned that students at "the mystic academy" play "skyball"? If so, this might just be a thinly disguised version of Harry Potter, and the whole novel is still lurking in the model's weights, ready to emerge. Maybe it will still write fiction that imitates the stylistic features of the Harry Potter novels. Both of these are the kind of things that copyright law could treat as infringing similarities, since similarity is not confined literally to the text.

**Questions:**

Can you provide more technical details? The descriptions in the paper are extremely concise and I am not convinced that this work is replicable based on those descriptions alone.

---

### Official Review · Reviewer_SCdB · 2023-11-07

**Soundness:** 4 excellent
**Presentation:** 3 good
**Contribution:** 4 excellent
**Rating:** 8
**Confidence:** 3

**Summary:**

This paper proposes a novel approach to unlearn a subset of the training data from LLMs, without having to retrain it from scratch. The proposed approach is expected to work well on copyrighted content, such as fiction, which comprises of unique words, phrases, and idiosyncratic stylistic expressions that differentiate it from natural language. It involves a two step approach where first a reinforced model is generated by finetuning on the training subset to identify which logits are reinforced and second, the model is finetuned to replace idiosyncratic expressions in the target data with their generic counterparts to break the entity links for next word prediction. The paper shows convincing evidence that a finetuned Llama2-7b model is effectively able to unlearn the Harry Potter series.

**Strengths:**

-  The problem statement of unlearning copyrighted content in LLMs without retraining from scratch is very timely and significant.
- The focus on the specific case study of unlearning Harry Potter series and the discussion of learnings in arriving at the proposed solution approach while also discussing solutions that dont work is very useful for the readers.

**Weaknesses:**

- The main shortcoming of the proposed approach which the paper mentions is that the unlearning approach heavily depends on the fiction content having identifiable anchor terms that are different from natural language. This makes the approach harder to generalize across a variety of copyrighted content including textbooks.

**Questions:**

It would be useful to measure how sensitive the proposed approach is to presence of anchor terms in the target content and the repetition of this content in the base LLM.